# Waist-to-Height Ratio, Waist Circumference, and Body Mass Index in Relation to Full Cardiometabolic Risk in an Adult Population from Medellin, Colombia

**DOI:** 10.3390/jcm14072411

**Published:** 2025-04-01

**Authors:** Mariana Montoya Castillo, Wilson de Jesús Martínez Quiroz, Milton Fabian Suarez-Ortegón, Luis Felipe Higuita-Gutiérrez

**Affiliations:** 1Facultad de Medicina, Universidad Cooperativa de Colombia, Medellín 050012, Colombia; mariana.montoyacasti@campusucc.edu.co; 2Dirección de Gestión Clínica y Promoción y Prevención, Metrosalud, Medellín 050010, Colombia; saludpublica2@metrosalud.gov.co; 3Departamento de Alimentación y Nutrición, Facultad de Ciencias de la Salud, Pontificia Universidad Javeriana Seccional Cali, Cali 760031, Colombia

**Keywords:** BMI, waist-to-height ratio, waist circumference, cardiometabolic risk

## Abstract

**Background/Objectives:** Few studies have compared the associations of different adiposity markers with cardiometabolic risk factors in individuals without diabetes or cardiovascular disease (CVD), particularly in South America. Moreover, the associations with more severe cardiometabolic risk, defined by the simultaneous presence of altered glycemia, blood pressure, and dyslipidemia, remain unknown. We examined whether the waist-to-height ratio (W-HtR), waist circumference (WC), and BMI were independently associated with cardiometabolic risk in a chronic disease prevention program in Medellín, Colombia. **Methods:** A cross-sectional study was conducted in 29,236 adults (age: 19–121 years) without diabetes or CVD. Exposures included increased W-HtR (>0.5), increased WC (≥80 cm for women, ≥90 cm for men), and overweight/obesity. The outcomes were dyslipidemia, elevated glycemia, high blood pressure, and full cardiometabolic risk (FCMR), defined as the presence of all three factors. Logistic regressions adjusted for sociodemographic and lifestyle covariates and additional adiposity markers were used. Cubic spline analyses examined the shape of associations. **Results:** Most individuals were over 40 years old (97.6%), only 40 were ≥100 years, and 16.5% (*n* = 4821) had FCMR. Increased W-HtR tripled the odds of FCMR compared with normal W-HtR (OR: 3.04, 95%CI: 2.45–3.77, *p* < 0.001). Increased WC doubled the odds of FCMR (*p* < 0.001). W-HtR remained the strongest predictor after adjusting for WC (OR: 1.99, 95%CI: 1.59–2.50) and BMI (OR: 2.48, 95%CI: 1.99–3.08). Cubic spline analyses showed a linear association between W-HtR and FCMR, whereas the BMI–FCMR association plateaued at approximately 30 kg/m^2^. **Conclusions:** In this cross-sectional study of a large middle-to-older-aged cohort, W-HtR was the strongest adiposity marker correlated with cardiometabolic risk.

## 1. Introduction

Cardiometabolic risk refers to the likelihood of developing cardiovascular diseases like heart attacks or strokes, as well as metabolic disorders, such as metabolic syndrome and diabetes, owing to the presence of certain risk factors. Traditional risk factors include arterial hypertension, hyperglycemia, and dyslipidemia [1]. Each of these three alterations can lead to the development of cardiovascular disease through different mechanisms. High blood pressure increases the afterload, producing ventricular hypertrophy, leading to cardiac dysfunction and multiple complications. It also produces endothelial dysfunction, affecting the microcirculation of all organs, including the heart and the brain [2]. Hyperglycemia is associated with the accumulation of advanced glycation end products in blood vessels, promoting inflammation and arterial obstruction. It also produces oxidative stress by generating free radicals that damage endothelial cells and cause chronic inflammation [3]. Finally, dyslipidemia mainly produces the formation of atheromatous plaque at the arterial level, which is perhaps the most important mechanism involved in the development of cardiovascular disease [4].

Additionally, within cardiovascular diseases, we also find cardiovascular–renal–hepatic–metabolic syndrome, which, as its name indicates, also involves a third and fourth organs, the kidney and the liver; this is because there are interconnected roles between these, and in the presence of metabolic syndrome, we can find dysfunction of one or all of these systems [5].

However, emerging factors such as inflammation measured by C-reactive protein, insulin resistance, psychosocial stress, and obesity are also gaining prominence. Among these factors, obesity stands out due to its epidemic proportions and its status as a modifiable risk factor.

According to the World Health Organization (WHO), in 2016, more than 1.9 billion adults were overweight, with over 650 million classified as obese. This represents approximately 39% of the world’s adult population being overweight and 13% being obese [6]. Recent data from a systematic review published in 2019 revealed high obesity rates across South America, with prevalence in adults ranging from 15% to 30% in various countries [7].

Given the significance of obesity and adjacent adiposity, various clinical and anthropometric indicators have been proposed for their monitoring. Among these prominent indicators are BMI (body mass index), WC (waist circumference), and W-HtR (waist-to-height ratio).

W-HtR and WC directly measure abdominal fat, which is more closely associated with metabolic and cardiovascular risk than overall body fat. Central obesity is linked to higher risks of diabetes, hypertension, and cardiovascular diseases, making these measures more predictive of health outcomes [8,9]. Measuring WC is straightforward and requires minimal equipment, typically a tape measure. This makes it easier to use in clinical and non-clinical settings. Also, W-HtR is calculated by simply dividing the waist circumference by the height, making it a quick and easy assessment.

The available evidence, coupled with the simplicity of these indicators and their ease of incorporation into clinical practice, has positioned them as predictors of cardiometabolic disease. In this regard, a systematic review comparing BMI, WC, and W-HtR found that W-HtR was a more accurate predictor of diabetes, hypertension, dyslipidemia, and metabolic syndrome in both males and females. In the pooled analysis evaluating all outcomes, W-HtR demonstrated an area under the ROC curve (AUC) of 0.704 for men and 0.725 for women, whereas BMI and WC had an AUC value lower than this threshold [8]. Another study conducted in Japan among adults concluded that W-HtR was a more accurate measure of central obesity and metabolic risk compared to BMI and WC, particularly in predicting the risk of metabolic syndrome [10]. Similarly, research conducted in the United Kingdom found that W-HtR surpassed BMI and WC in predicting health risks, including hypertension, diabetes, and cardiovascular diseases [11].

In Latin America, there is scarce evidence of evaluation in middle-aged and elderly population, without cardiovascular disease or diabetes, in addition to the association of total cardiometabolic risk (presence of increased glycemia, dyslipidemia and hypertension) with the different methods of adiposity determination (BMI, waist circumference, and W-HtR with cardiometabolic risk) and which of them has the best performance. In 2016, Márcia Mara and colleagues conducted a study in Brazil, concluding that W-HtR is a highly effective screening tool for identifying various health risks. Its superiority over BMI and WC in predicting conditions such as cardiovascular disease, diabetes, hypertension, and metabolic syndrome makes it a valuable addition to clinical practice. The authors advocate for the broader adoption of W-HtR in health assessments to improve the early detection and prevention of cardiometabolic diseases [12]. Another study conducted in Chile in a prospective cohort found that W-HtR was the best way to predict cardiovascular risk factors and all-cause mortality [13].

Particularly in Colombia, the need to estimate this association is crucial, given its status as a low-middle income country where cardiovascular diseases are the leading cause of mortality, with a 12% increase observed in 2019 compared to 2009 [14]. It should also be considered that in Colombia, according to the national nutritional situation survey in 2015, 56% of the population had obesity. Therefore, it is pertinent to compare different markers of adiposity in relation to cardiometabolic risk in a population as particular as the Colombian population.

Therefore, we conducted a study in an adult population in Medellín, Colombia, to determine the association of BMI, waist circumference, and W-HtR with the complete cardiometabolic risk (defined by the presence of three events: dyslipidemia, hyperglycemia, and increased blood pressure).

## 2. Methods

### 2.1. Study Population

This cross-sectional study analyzed data from individuals participating in a public health program focused on managing chronic diseases, administered by E.S.E Metrosalud in Medellín, Ant., Colombia in 2019. The initial dataset included 69,883 records of individuals aged 18 years or older, with data extracted from electronic medical records. The study included adults (18 years and older) who actively participated and adhered to the program screenings. Cases with missing data or extreme values for anthropometric and cardiometabolic risk variables were excluded. The criteria used to identify outliers are detailed in Appendix A. Because the study targeted individuals at risk of cardiometabolic disease rather than those already diagnosed with such conditions, individuals with diabetes (ICD-10 CODES E-110-149; E-100-109), kidney disease (ICD-10 CODES N170-179), cerebrovascular disease (ICD-10 I600-679; G-450-459), and cardiovascular disease (ICD-10 I200-I259) were excluded. In addition, cases with missing sociodemographic and lifestyle covariates, which were necessary for adjusting associations between anthropometric markers and cardiometabolic risk, were removed. The final sample size for analysis, consisting of 29,236 individuals, is detailed in Appendix A.

### 2.2. Adiposity Markers—Independent Variables

The study assessed adiposity using body mass index (BMI), waist circumference (WC), and waist-to-height ratio (W-HtR), all derived from weight and height measurements taken using standard procedures. BMI was calculated as weight (kg) divided by height squared (m^2^), while W-HtR was determined by dividing waist circumference by height. The study categorized increased WC as ≥80 cm for women and ≥90 cm for men, following the recommended thresholds for abdominal obesity in South American populations [15]. Increased W-HtR was defined as values greater than 0.5 [12], while overweight and obesity were classified based on BMI values of ≥25 and ≥30, respectively [6].

### 2.3. Cardiometabolic Risk—Dependent Variables

Three cardiometabolic components were assessed: elevated blood glucose, dyslipidemia, and high blood pressure. These were determined using criteria from the harmonized definition of metabolic syndrome [15] and medical diagnosis records. Elevated blood glucose was defined as glucose levels ≥ 100 mg/dL. Dyslipidemia was identified by triglyceride levels exceeding 150 mg/dL, low high-density lipoprotein cholesterol (HDL-C) levels (<40 mg/dL in men and <50 mg/dL in women), low-density lipoprotein cholesterol (LDL-C) levels above 110 mg/dL (due to its link to atherogenesis and cardiovascular risk), or a clinical diagnosis of dyslipidemia. High blood pressure was defined as systolic blood pressure (SBP) ≥ 130 mmHg and/or diastolic blood pressure (DBP) ≥ 85 mmHg or a diagnosis of hypertension. Full cardiometabolic risk was defined as the presence of all three components.

Blood glucose, triglycerides, and cholesterol were measured using the Cobas 6000 analyzer (Roche, Basel, Switzerland) via a colorimetric enzymatic test. Blood pressure was measured manually or automatically with calibrated equipment.

### 2.4. Sociodemographic and Lifestyle Covariates

Several sociodemographic and lifestyle factors were included as covariates to adjust associations in the analysis. These included age, sex (female as the reference category), education level (illiterate [reference category], elementary, secondary, technical or vocational education, undergraduate, and graduate), ethnicity (general population [reference category], Afro-descendant, Palenquero, Raizal, Roma, and Indigenous), area of residence (urban [reference category], rural), and marital status (single [reference category], divorced, cohabiting, married, or widowed). Alcohol consumption (yes/no) and smoking (yes/no) were also considered. Ethnicity was self-reported based on personal identification with an ethnic group. Smoking and alcohol consumption were determined based on self-reported use at least once a week over the past month. Physical activity was classified according to the World Health Organization (WHO) definition: individuals engaging in moderate to vigorous physical activity at least three times per week were considered active, while those exercising less frequently were categorized as inactive [16].

### 2.5. Data Analysis

Descriptive statistics were used to summarize study variables. Continuous variables were reported as medians with interquartile ranges, while categorical variables were expressed as counts and percentages. Associations between adiposity markers (increased W-HtR, increased WC, and BMI categories) and each cardiometabolic component were analyzed using logistic regression models, both unadjusted and adjusted for the covariates mentioned (adjustment model 1).

To assess the association between adiposity markers and full cardiometabolic risk (the presence of all three risk components), additional models were adjusted separately for each adiposity marker. For example, the association between increased W-HtR and full cardiometabolic risk was analyzed unadjusted, then adjusted for model 1, followed by adjustments including WC and BMI in separate models. These additional adjustments aimed to determine whether associations remained independent of other adiposity markers.

Restricted cubic spline analyses were conducted to explore the relationship between adiposity markers (as continuous variables) and full cardiometabolic risk, adjusting for covariates (model 1) with four knots at the 5th, 35th, 65th, and 95th percentiles of adiposity markers. Interaction terms between sex and adiposity markers were also tested to determine whether stratified analyses by sex were necessary.

All analyses were conducted using STATA 14.2 software.

## 3. Results

Nearly all the study sample participants were aged above 40 years old (only 2.4% were younger than this threshold), and there were almost three times the number of women than men (Table 1). Medians of BMI, WC, and W-HtR were in ranges of increased values for these variables (Table 1). Most of the individuals of the study ethnically self-perceived as general population (not ethnic minority groups), were single or married, resided in the urban area of Medellin, and were physically inactive (Table 1).

There was a high prevalence of increased W-HtR, increased WC, and overweight/obesity: 95%, 88.2%, and 40.8%/33.8%, respectively. Despite the very large proportion of people with increased W-HtR (W-HtR > 0.5), the median of W-HtR indicated that 50% of the individuals were below 0.6 and the remaining 50% were above this value.

The *p* values for the interaction between sex and increased values of W-HtR, WC, and overweight/obesity in relation to full cardiometabolic risk were 0.275, 0.371, and 0.293/0.194, respectively. Therefore, the association analyses are further described for the whole sample, without discriminating by sex and adjusting for sex along with the other covariates.

Dyslipidemia and increased blood pressure were highly prevalent, with more than two-third parts of the individuals having these cardiometabolic components (Table 2). A total of 91.5% of the subjects had at least one cardiometabolic component, and 16.5% presented all three kind of components (full cardiometabolic risk) (Table 2).

When analyzing specific cardiometabolic risk components, both before and after adjusting for various demographic and lifestyle factors, an increased W-HtR consistently demonstrated a stronger and more positive association with each cardiometabolic risk factor compared to an increased WC. However, WC was also significantly associated with these risk factors (Table 3). On the other hand, BMI, particularly in the categories of overweight and obesity, was also significantly associated with an increased risk of all three metabolic disturbances. However, the strength of these associations tended to be slightly weaker compared to W-HtR and WC (Table 3). The most important association of cardiometabolic risk was dyslipidemia when the patient had W-HtR > 0.5 in the adjusted model, increasing the opportunity in almost two times (Table 3).

Table 4 describes the odds ratios for association between each adiposity marker and full cardiometabolic risk. In terms of unadjusted and adjustment for sociodemographic and lifestyle covariates, individuals with an increased W-HtR had three times the opportunity to present full cardiometabolic risk of those with normal W-HtR. Having an increased WC doubled the opportunity of presenting full cardiometabolic risk compared with having a normal WC. A similar pattern of doubled odds for full cardiometabolic risk was observed for obesity vs. normal weight (Table 4). When the above associations were further adjusted for the other adiposity markers, increased W-HtR persisted as the marker most strongly significantly associated with full cardiometabolic risk with odds ratios around 2.0 in models additionally adjusted for WC and BMI. The statistical significance of the associations of increased WC and obesity with full cardiometabolic risk remained after further adjustment for the other adiposity markers, and these adjustments reduced the strength of those associations in 15–22% and 22–26% for increased WC and obesity, respectively.

The shape of the associations between each adiposity marker and the full cardiometabolic risk is depicted in Figure 1. The association between W-HtR and full cardiometabolic risk becomes evident at W-HtR values of approximately 0.6. Similarly, full cardiometabolic risk is strongly associated with WC values around 100 cm. Odds ratios for full cardiometabolic risk was to some extent more linearly proportional to values of W-HtR (particularly from a value of 0.7 approximately) than to WC values (Figure 1). From BMI values close to the threshold of obesity (30 Kg/m^2^), the association between this marker and full cardiometabolic risk is more evident but with a flat pattern in which the odds ratios for full cardiometabolic risk remain constant despite increasing BMI values (Figure 1).

## 4. Discussion

Our study showed that obesity measured by W-HtR has a greater association with cardiometabolic outcomes, such as dyslipidemia, increased glycemia, increased blood pressure, and complete cardiometabolic risk, consisting of the presence of the three aforementioned outcomes. These results are consistent with the findings in the systematic review published in 2012 [8], and also in other studies conducted in several countries such as Japan [11], Brazil [12], and the United Kingdom [13]. Table 5 illustrates some of the most recent studies, describing the main results. It is important to clarify that none of the studies evaluated the presence of the three outcomes together, and in most of them, only one outcome was evaluated.

The association between a higher W-HtR and increased cardiometabolic risk is biologically plausible and well-supported by the pathophysiology of fat distribution, particularly central obesity. This type of obesity, indicated by a higher W-HtR, reflects an accumulation of visceral fat, and visceral adipose tissue is more metabolically active than subcutaneous fat, producing a variety of adipokines and cytokines that influence metabolic processes [22,23]. These substances, including interleukin-6 (IL-6) and tumor necrosis factor-alpha (TNF-α), contribute to a state of chronic low-grade inflammation, a hallmark of metabolic syndrome and cardiovascular disease [24,25]. The release of free fatty acids from visceral fat directly into the portal vein, which transports blood to the liver, can lead to hepatic insulin resistance. This disrupts glucose metabolism and promotes the development of type 2 diabetes. Elevated W-HtR, reflecting greater visceral fat, correlates strongly with increased insulin resistance [26,27]. Visceral fat is also linked to an adverse lipid profile characterized by high levels of triglycerides and low levels of high-density lipoprotein (HDL) cholesterol. These changes are significant risk factors for atherosclerosis and cardiovascular disease [28]. Finally, adipokines produced by visceral fat, such as angiotensinogen, contribute to the regulation of blood pressure. Increased visceral fat can lead to elevated levels of angiotensin II, promoting hypertension [29].

In this study, W-HtR emerged as the adiposity marker most strongly associated with cardiometabolic risk, surpassing both BMI and WC. W-HtR has been increasingly recognized as a superior screening tool for assessing cardiometabolic risk compared with BMI. This advantage is primarily due to W-HtR’s ability to reflect central obesity more accurately, which is closely linked to metabolic and cardiovascular diseases, as we mentioned above [30]. Numerous studies have demonstrated that W-HtR is superior predictive power for cardiovascular diseases. A meta-analysis by Lee et al. found that W-HtR was a better discriminator of cardiovascular risk factors than BMI, highlighting its stronger association with blood pressure, cholesterol levels, and other cardiovascular indicators [31].

On the other hand, W-HtR is straightforward to measure and calculate, requiring only two measurements: WC and height. This simplicity reduces the likelihood of measurement errors and makes it feasible for large-scale screenings and clinical settings. In contrast, BMI requires accurate weight and height measurements and the use of a formula, which can be more cumbersome [32]. Additionally, W-HtR is applicable across different age, sex, and ethnic groups without the need for adjustments or specific thresholds. This universality simplifies its use in diverse populations, ensuring consistency and comparability of results. BMI categories (underweight, normal weight, overweight, and obese) often require age- and sex-specific adjustments, which can complicate their application [33].

In our study, we found that the relationship between W-HtR and full cardiometabolic risk becomes evident at a W-HtR value of approximately 0.6. A W-HtR of 0.6 or higher is associated with an increased risk of cardiometabolic conditions. Recent clinical studies have indicated that a W-HtR greater than 0.6, as found in our study, is significantly associated with an elevated cardiometabolic risk, surpassing the traditionally used threshold of 0.5. The research highlights that individuals with a W-HtR over 0.6 exhibit higher incidences of hypertension, type 2 diabetes, and dyslipidemia, suggesting that the higher cut-off point better identifies those at greater risk for cardiometabolic conditions and also improves the specificity [8,34,35].

### Weakness and Strengths

The most important limitations of our study are that it was developed only in E.S.E Metrosalud and most of the population was over 40 years old, however, the main strength of the study is that it is the first one carried out in the city of Medellin-Colombia. Also, a robust sample size was used and some of the most important cardiometabolic factors, such as diabetes, arterial hypertension, and dyslipidemia, were evaluated.

Another important limitation is that this was a cross-sectional study, which does not allow for the identification of causal relationships. Similarly, there could be potential residual confounding, and due to variability in W-HtR across different populations, our findings may not be generalizable until they are replicated in other populations. Finally, it is important to note that the intensity of physical activity was not assessed using objective scales such as the IPAQ, but rather through self-report.

## 5. Conclusions

In conclusion, our study found that W-HtR was the adiposity marker most strongly correlated with cardiometabolic risk. Unlike other markers such as Body Mass Index (BMI) and Waist Circumference (WC), W-HtR exhibited a more linear and proportional association with cardiometabolic risk factors. However, prospective studies are needed to validate our cross-sectional findings. Given the potential influence of W-HtR variability, these associations should be evaluated in different geographical and ethnic populations.

## Figures and Tables

**Figure 1 jcm-14-02411-f001:**
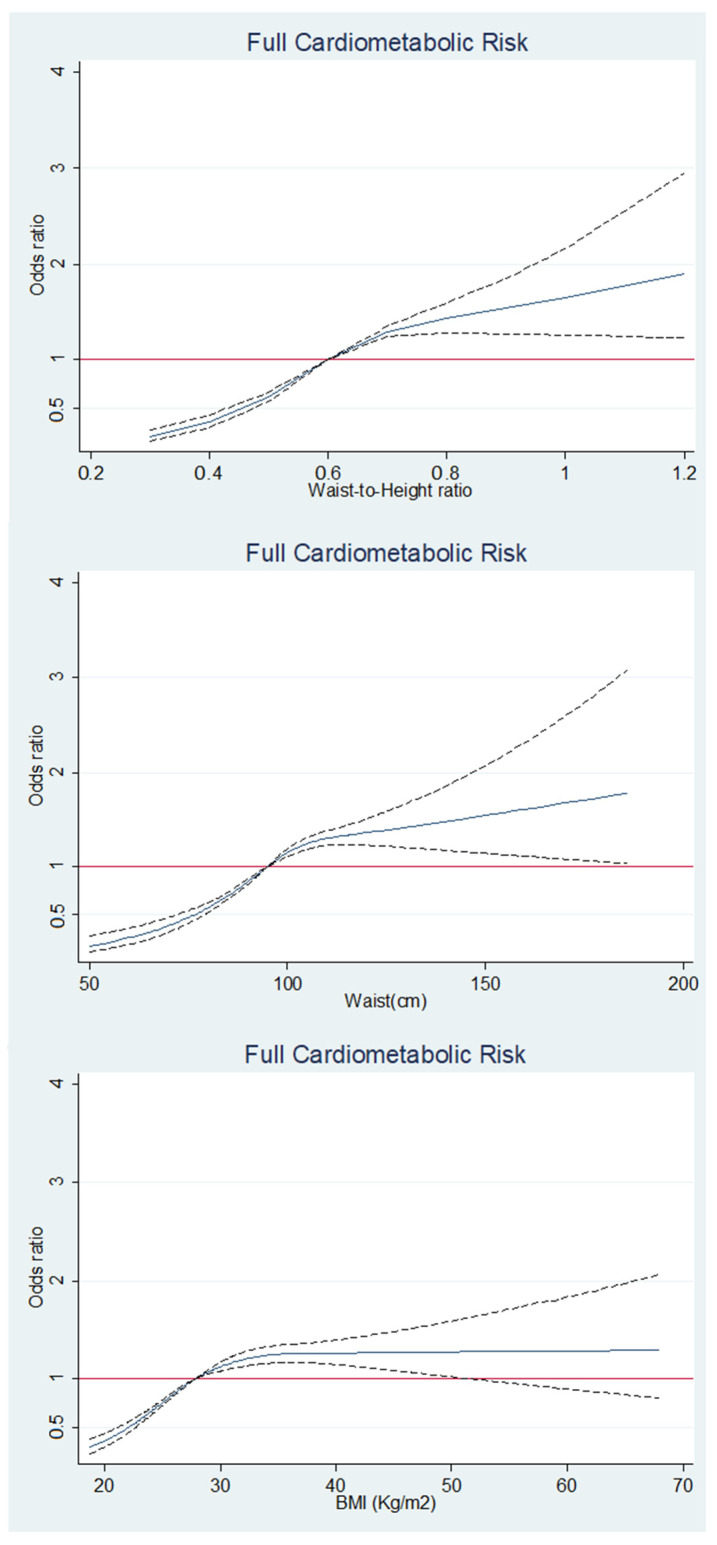
Cubic splines analyses for the waist to height ratio, waist circumference, and BMI. Solid Blue Line: Represents the estimated odds ratio for full cardiometabolic risk across different values of the variable. Dashed Black Lines: Indicate the 95% confidence interval for the estimated odds ratio. Solid Red Line: Serves as a reference line at an odds ratio of 1, indicating no association.

**Table 1 jcm-14-02411-t001:** Description of study variables in the sample of individuals (*n* = 29,236).

Age Group (Years)	*n* (%)
18–39	720 (2.4)
40–60	7743 (26.5)
>60	20,723 (71)
Sex Female *n* (%)	21,410 (73.2)
Clinical, anthropometrical, and biochemical variables *	
BMI (Kg/mt^2^)	27.9 (25–31.5)
Waist circumference (WC) cm	95 (88–102)
Waist-to-Height Ratio (W-HtR)	0.6 (0.6–0.7)
Systolic blood pressure mmHg	125 (120–140)
Diastolic blood pressure mmHg	80 (70–80)
Glucose mg/dL	96 (90–105)
Triglycerides mg/dL	150 (111–206)
HDL cholesterol mg/dL	44 (37–52)
LDL cholesterol mg/dL	110 (86–135)
Covariates	
Education level *n* (%)	
Illiterate	4432 (15.1)
Elemental	17,537 (59.9)
Secondary	6850 (23.4)
Technical education	263 (0.9)
Undergraduate	125 (125)
Graduate	29 (0.1)
Marital status *n* (%)	
Single	11,566 (39.5)
Divorced	1381 (4.5)
Free union	3193 (10.9)
Married	10,358 (35.4)
Widow/Widower	2788 (9.5)
Ethnicity *n* (%)	
General population	25,995 (88.9)
Afrodescendant	725 (2.5)
Indígenous	122 (0.4)
Palenquero	96 (0.3)
Raizal	2180 (7.4)
ROM	118 (0.4)
Residential area *n* (%)	
Urban	27,332 (93.5)
Rural	1904 (6.5)
Alcohol consumption *n* (%)	
Yes	1314 (4.5)
No	27,922 (95.5)
Smoking *n* (%)	
Yes	3412 (11.7)
No	25,824 (88.3)
Physical activity *n* (%)	
Yes	8810 (30.1)
No	20,426 (69.9)

* Data are medians (interquartile range).

**Table 2 jcm-14-02411-t002:** Cardiometabolic risk components and sub-components in the sample of individuals (*n* = 29,236).

*n*	(%)
Increased glycemia Component High fasting glucose (≥100 mg/dL)	11,802 (37.9)
Dyslipidemia component	18,294 (62.6)
High triglyceride level (≥150 mg/dL)	14,493 (49.6)
Low HDL-C	17,679 (60.5)
Increased LDL-C (>110 mg/dL)	14,790 (50.6)
Diagnosis of dyslipidemia	325 (1.2)
Increased Blood pressure component	19,229 (65.7)
(SBP ≥ 130 or DBP ≥ 85 mg/dL)	14,570 (49.9)
Diagnosis of hypertension	9821 (35.6)
Number of Cardiometabolic risk components	
0	2494 (8.5)
1	9700 (33.2)
2	12,221 (41.8)
3	4821 (16.5)

**Table 3 jcm-14-02411-t003:** Odds ratios (OR) and 95% confidence interval (95%CI) for each kind of cardiometabolic risk component by international cut-off points of W-HtR, WC, and BMI.

	Increased Glycemia Component		Increased Blood Pressure Component		Dyslipidemia Component	
	OR (95%CI)	OR (95%CI)	OR (95%CI)	OR (95%CI)	OR (95%CI)	OR (95%CI)
	Unadjusted	Adjusted for Model 1 *	Unadjusted	Adjusted for Model 1 *	Unadjusted	Adjusted for Model 1 *
W-HtR						
≤0.5						
>0.5 (increased)	1.74 (1.53–1.80)	1.70 (1.51–1.92)	1.20 (1.08–1.34)	1.23 (1.10–1.38)	1.78 (1.60–1.98)	1.76 (1.58–1.96)
*p* value	<0.001	<0.001	0.001	<0.001	<0.001	<0.001
WC						
Normal WC						
Increased WC	1.66 (1.53–1.80)	1.73 (1.60–1.88)	1.08 (1.006–1.16)	1.18 (1.10–1.28)	1.64 (1.52–1.76)	1.58 (1.46–1.70)
*p* value	<0.001	<0.001	0.033	<0.001	<0.001	<0.001
BMI						
Normal weight						
Overweight	1.34 (1.26–1.42)	1.39 (1.30–1.48)	1.08 (1.02–1.15)	1.13 (1.06–1.20)	1.46 (1.38–1.55)	1.43 (1.34–1.52)
*p* value	<0.001	<0.001	0.007	<0.001	<0.001	<0.001
Obesity	1.73 (1.62–1.84)	1.90 (1.78–2.03)	1.18 (1.11–1.26)	1.32 (1.24–1.41)	1.47 (1.38–1.57)	1.37 (1.28–1.46)
*p* value	<0.001	<0.001	<0.001	<0.001	<0.001	<0.001

* Adjusted for age (years), sex (male/female [Ref]), education level (illiterate [Ref], elementary, secondary, technical profession education, undergraduate, graduate), ethnicity (general population [Ref], Afro-descendant, Palenquero, Raizal, Roma, and Indigenous) area of residence (urban [Ref]/rural), marital status (single [Ref], divorced, free union, married, widowhood), alcohol intake habit (yes/no), and smoking (yes/no).

**Table 4 jcm-14-02411-t004:** Odds ratios (OR) and 95% confidence interval (95%CI) for full cardiometabolic risk by international cut-off points of W-HtR, WC, and BMI.

	OR (95%CI)	*p* Value	OR (95%CI)	*p* Value	OR (95%CI)	*p* Value	OR (95%CI)	*p* Value
	Unadjusted		Adjusted for Model 1 *		Adjusted for Model 1 Plus WC		Adjusted for Model 1 Plus BMI	
W-HtR								
≤0.5	1.0 (Reference)		1.0 (Reference)		1.0 (Reference)		1.0 (Reference)	
>0.5	3.09 (2.49–3.83)	<0.001	3.04 (2.45–3.77)	<0.001	1.99 (1.59–2.50)	<0.001	2.48 (1.99–3.08)	<0.001
					Adjusted for model 1 plus W-HtR		Adjusted for model 1 plus BMI	
WC								
Normal WC	1.0 (Reference)		1.0 (Reference)		1.0 (Reference)		1.0 (Reference)	
Increased WC	1.95 (1.73–2.20)	<0.001	2.04 (1.80–2.30)	<0.001	1.55 (1.36–1.77)	<0.001	1.70 (1.50–1.93)	<0.001
					Adjusted for model 1 plus WC		Adjusted for model 1 plus W-HtR	
BMI								
Normal weight	1.0 (Reference)		1.0 (Reference)		1.0 (Reference)		1.0 (Reference)	
Overweight	1.57 (1.44–1.71)	<0.001	1.61 (1.48–1.76)	<0.001	1.42 (1.30–1.56)	<0.001	1.46 (1.34–1.60)	<0.001
Obesity	1.88 (1.72–2.05)	<0.001	2.01 (1.83–2.20)	<0.001	1.48 (1.32–1.65)	<0.001	1.59 (1.42–1.77)	<0.001

* Adjusted for age (years), sex (male/female [Ref]), education level (illiterate [Ref], elementary, secondary, technical profession education, undergraduate, graduate), ethnicity (general population [Ref], Afro-descendant, Palenquero, Raizal, Roma, and Indigenous) area of residence (urban [Ref]/rural), marital status (single [Ref], divorced, free union, married, widowhood), alcohol intake habit (yes/no), and smoking (yes/no).

**Table 5 jcm-14-02411-t005:** Selected studies on waist-to-height ratio and cardiometabolic risk.

Authors, Year, Ref.	Country	Design	*n* (% Male), Age	Exposure (s); Outcome	Results
Nguyen Ngoc et al., 2019, [17]	Thailand	Cross-sectionalsurvey	15,842 (47.4%), 59.3 ± 13.2 years	W-HtR, WC, BMI; hypertension	Regardless of gender, the best method to distinguish performance in predict arterial hypertension was waist-to-height ratio (W-HtR) [AUC: 0.640 (0.631–0.649)]
Liu et al., 2019, [18]	China	Prospective cohort study	4416 (41.2%), >65 years	W-HtR, WC, BMI; dyslipidemia, hypertension, hyperglycemia	Compared with other anthropometric indices, W-HtR had significantly higher areas under the curve (AUCs) for predicting dyslipidemia (AUCs: 0.646, sensitivity: 65%, specificity: 44%), hyperglycemia (AUCs: 0.595, sensitivity: 60%, specificity: 45%), and CVDs (AUCs: 0.619, sensitivity: 59%, specificity: 41%)
Rodríguez Guerrero et al., 2020, [19]	Spain	Cross-sectional study	361 (46.8%), 73.2 ± 6.4 years	W-HtR, WC, BMI; metabolic syndrome	The W-HtR and the basal glucose had the best predictive capacity (S = 61.4%, SP = 89.2%, PPV = 81.5, validity index or VI = 77%)
Alves et al., 2021, [20]	Brazil	Cross-sectional study	159 (49.7%), 70.9 ± 7.4 years	W-HtR, WC, BMI; metabolic syndrome	Lipid accumulation product (LAP) and W-HtR resulted in the largest AUC values (>0.80). In both sexes, the best indicators were LAP, WC, and WHtR.
Marzban et al., 2022, [21]	Iran	Prospective cohort study	3000 (48.5%), 67.75 ± 7.1 years	W-HtR, WC, BMI; metabolic syndrome	The highest adjusted RRs for metabolic syndrome were observed for the following indices: W-HtR (RR = 15.24), Fat-to-muscle ratio (RR = 4.341), and Waist-to-hip ratio (RR = 3.14).

## Data Availability

The data presented in this study are available on request from the corresponding author.

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
