# Peer review of "Waist-to-Height Ratio, Waist Circumference, and Body Mass Index in Relation to Full Cardiometabolic Risk in an Adult Population from Medellin, Colombia"

_jcm, 2025, doi:10.3390/jcm14072411_

Round 1

Reviewer 1 Report

Comments and Suggestions for Authors

In this study by Montoya Castillo and colleagues titled “Waist-to-Height Ratio, Waist circumference, and Body Mass Index in relation to Full Cardiometabolic Risk in an adult population from Medellin, Colombia”, waist-to-height ratio (W-HtR), waist circumference (WC), and BMI and their independent association with cardiometabolic risk were investigated. Here are my comments related to this manuscript. 

-In the abstract section, please include the age range of the participants in this study. 

-There are some typos and words with grammatical errors throughout the manuscript.

-The references 7 and 8 are missing in the introduction section. 

-Introduction section could be improved by adding information on how dyslipidemia, hyperglycemia, and increased blood pressure are implicated in the development of cardiovascular diseases. 

-Supplementary materials are missing.

-Methods section needs improvement, for example, it is not clear how dyslipidemia, hyperglycemia, and increased blood pressure were defined in this study. Please add the values range for each cardiometabolic component that was considered in this study. Also, clarify whether participants did not have metabolic syndrome. 

-The authors should justify why these values of the adiposity markers-independent variables were considered. 

-Please add more details in methods on the methods of defining physical activity (vigorous or moderate physical activity at least three times per week were classified as active). How was each category of physical activity determined?

-The data analysis section does not describe how the median and interquartile range were calculated. 

-Figure 1 has no P values ​​or confidence intervals. Please add this information.

Author Response

Medellín March 24nd, 2025

Journal of Clinical Medicine

MDPI

Subject: Answer to reviewers of the manuscript “Waist-to-Height Ratio, Waist circumference, and Body Mass Index in relation to Full Cardiometabolic Risk in an adult population from Medellin, Colombia”

Dear Editor,

Thank you for your guidance in reviewing our submission. The manuscript has been revised and the reviewers’ comments have been addressed below. We are thankful to the reviewers for their valuable suggestions for improving the manuscript, and we hope it is now acceptable for publication. Responses to each reviewer are included in yellow highlights within the manuscript.

Thank you for your consideration.

Sincerely,

The authors

Reviewer 1

Comment 1: In the abstract section, please include the age range of the participants in this study.

Answer: Age range included accordingly.  

Comment 2: There are some typos and words with grammatical errors throughout the manuscript.

Answer: Some words were corrected, but if you identify any errors, please let us know.

Comment 3: The references 7 and 8 are missing in the introduction section.

Answer: Reference 7 is in the introduction but is now numbered as 9, due to suggested changes in the introduction (highlighted in yellow). We had omitted reference 8, so we corrected the listing.

Comment 4: Introduction section could be improved by adding information on how dyslipidemia, hyperglycemia, and increased blood pressure are implicated in the development of cardiovascular diseases.

Answer: We added this information: Each of these three alterations can lead to the development of cardiovascular disease through different mechanisms. High blood pressure increases afterload, producing ventricular hypertrophy, leading to cardiac dysfunction and multiple complications. It also produces endothelial dysfunction, affecting the microcirculation of all organs, including the heart and the brain (2). Hyperglycemia is associated with the accumulation of advanced glycation end products in blood vessels, promoting inflammation and arterial obstruction. It also produces oxidative stress by generating free radicals that damage endothelial cells and cause chronic inflammation (3). Finally, dyslipidemia mainly produces the formation of atheromatous plaque at the arterial level, which is perhaps the most important mechanism involved in the development of cardiovascular disease (4).

Comment 5: Supplementary materials are missing.

Answer: Supplemental material was mistakenly missed in the first submission. Apologies for this. The items included in the supplementary material are as follows:

Table S1. Definition of potential extreme/implausible values for anthropometric and biochemical  variables

Values

Glucose (mg/dl)

<30 or >600

 Triglycerides (mg/dl)

<10 or >1.000

HDL-C (mg/dl)

<10 or >118

SBP (mm / Hg)

<80 or >280

DBP (mm/Hg)

<50 or  >195

Waist (cm)

<50 or >198

BMI (Kg/m2)

>70

SBP=Systolic Blood Pressure; DBP=Diastolic Blood Pressure; C=Cholesterol.

Table S2. Identification of the study sample

Initial sample

Exclusion criteria

n excluded

Remaining sample

69,883

Missing values for

anthropometric and biochemical  variables

19,505

50,378

50,378

Potential extreme/implausible values for

anthropometric and biochemical  variables

2,589

47,789

47,789

Diagnostic of  any of the following diseases: diabetes, kidney disease, cerebrovascular and cardiovascular disease

10,231

37,558

37,558

Missing values for covariates of age, sex, education level, ethnicity, area of residence, marital status, alcohol consumption, and smoking

8,322

29,236

(Final sample)

Comment 6: Methods section needs improvement, for example, it is not clear how dyslipidemia, hyperglycemia, and increased blood pressure were defined in this study. Please add the values range for each cardiometabolic component that was considered in this study. Also, clarify whether participants did not have metabolic syndrome.

Answer: In the section on cardiometabolic risk dependent variables in methods, the values ​are defined: Increased glycemia: having glucose levels ≥100 mg/dl. Dyslipidemia: either having triglycerides >150 mg/dl, or high-density lipoprotein cholesterol (HDL-C) <40 mg/dl in men and <50 mg/dl in women, or low-density lipoprotein cholesterol (LDL-C) levels >110 mg/dL (additionally considered due its relationship with atherogenesis and cardiovascular risk), or current diagnosis of dyslipidemia.  Increased blood pressure: having a systolic blood pressure (SBP) ≥130 mm/Hg and/or diastolic blood pressure (DBP) ≥85 mm/Hg, or diagnosis of hypertension.

Additionally, in the study population section in methods, it is clarified that the exclusion criteria include: individuals diagnosed with diabetes (ICD-10 CODES E-110-149; E-100-109), kidney disease (ICD-10 CODES N170-179), cerebrovascular (ICD-10 I600-679; G-450-459), and cardiovascular disease (ICD-10 I200-I259). We did not want to exclude metabolic syndrome since, on the contrary, we wanted to evaluate the association of anthropometric measurements with complete cardiovascular risk.

Comment 7: The authors should justify why these values of the adiposity markers-independent variables were considered.

Answer: We have inserted references for each cut-off used:

The study categorized increased WC as ≥80 cm for women and ≥90 cm for men, following the recommended thresholds for abdominal obesity in South American populations (15). Increased W-HtR was defined as values greater than 0.5 (12), while overweight and obesity were classified based on BMI values of ≥25 and ≥30, respectively (6).”

Comment 8: Please add more details in methods on the methods of defining physical activity (vigorous or moderate physical activity at least three times per week were classified as active). How was each category of physical activity determined?

Answer: The determination of physical activity intensity was conducted during the medical consultation by asking the patient whether they engage in moderate to vigorous physical activity for at least three times per week. For this reason, we noted in the limitations that physical activity was not assessed using objective scales, such as the IPAQ, but rather through self-report.

Comment 9: The data analysis section does not describe how the median and interquartile range were calculated.

Answer: At the end of the data analysis section it is specified that all analyses were conducted using STATA 14.2 software, including median and interquartile range. Interquartile range always consist of 25th and 75yh percentiles. This descriptive parameter is a central trend measure based on position of values (from lowest to highest) of the variable, and thus it does not exist a calculation for this.

Comment 10: Figure 1 has no P values ​​or confidence intervals. Please add this information.

Answer: Since this is a cubic spline analysis, we cannot demonstrate a specific confidence interval since it varies depending on the point within the domain of the independent variable.

Reviewer 2

Comment 1: The introduction provides a solid background on cardiometabolic risk factors and their relationship with adiposity markers. It discusses the significance of BMI, waist circumference (WC), and waist-to-height ratio (W-HtR), citing previous studies. However, while references to global research are included, there is limited discussion of prior studies conducted in similar Latin American populations. Expanding on regional studies would improve contextual relevance.

Answer: We added to the introduction one more study found in Latin America about the importance of W-HtR:

Another study conducted in Chile in a prospective cohort found that W-HtR was the best way to predict cardiovascular risk factors and all-cause mortality (13).”

Comment 2: The cross-sectional study design is appropriate for examining associations between adiposity markers and cardiometabolic risk. The large sample size (n=29,236) strengthens the reliability of the results. However, being cross-sectional, it does not allow causal inferences. The methods section is detailed and describes how data were collected, including the exclusion criteria, measurement techniques, and statistical analyses. However, a more explicit discussion of how missing data were handled and how potential confounders were selected would enhance clarity.

Answer: In the supplementary material, we include the following table, in which we present the number of patients excluded due to missing data.

Table S2. Identification of the study sample

Initial sample

Exclusion criteria

n excluded

Remaining sample

69,883

Missing values for

anthropometric and biochemical  variables

19,505

50,378

50,378

Potential extreme/implausible values for

anthropometric and biochemical  variables

2,589

47,789

47,789

Diagnostic of  any of the following diseases: diabetes, kidney disease, cerebrovascular and cardiovascular disease

10,231

37,558

37,558

Missing values for covariates of age, sex, education level, ethnicity, area of residence, marital status, alcohol consumption, and smoking

8,322

29,236

(Final sample)

Comment 3: The results are well-structured with clear tables and figures. The logistic regression models and cubic spline analyses are well explained. However, the readability of some sections could be improved by summarizing key findings before presenting detailed statistical outputs.

Answer: We supplement the results with some summarized information before showing the statistical analyses.

Comment 4: The conclusions accurately reflect the findings, emphasizing the strong predictive role of W-HtR in cardiometabolic risk. However, the authors suggest W-HtR should be the preferred screening tool without adequately acknowledging potential limitations such as variability across different populations.

Answer: We added some limitations in the weakness and conclusions sections:

“Another important limitation is that this was a cross-sectional study, which does not allow for the identification of causal relationships. Similarly, there could be potential residual confounding, and due to variability in W-HtR across different populations, our findings may not be generalizable until they are replicated in other populations.”

Reviewer 3

Comment 1: The authors concluded: “In this large middle-to-older-aged cohort, W-HtR was the strongest adiposity marker linked to cardiometabolic risk, with a more linear association pattern”. However, given the cross-sectional design of the study, results on such association or prediction could be misleading (the study design does not allow for causal inference)

Answer: We have moderated the conclusion in abstract and end of the discussion:

Abstract:

“In this cross-sectional study of a large middle-to-older-aged cohort, W-HtR was the strongest adiposity marker correlated to cardiometabolic risk.”

Discussion:

“In conclusion, our study found that W-HtR was the adiposity marker most strongly correlated with cardiometabolic risk. Unlike other markers such as Body Mass Index (BMI) and Waist Circumference (WC), W-HtR exhibited a more linear and proportional association with cardiometabolic risk factors. However, prospective studies are needed to validate our cross-sectional findings. Given the potential influence of W-HtR variability, these associations should be evaluated in different geographical and ethnic populations.”

Comment 2: Given the fact that it represents a novel paper on cardiovascular and metabolic factors, it would be useful to discuss the (relative) new cardiovascular-kidney-metabolic (CKM) syndrome

Answer: We added reference 5, which has as its central theme the Cardiovascular-Renal-Hepatic-Metabolic (CRHM) syndrome

Comment 3: Also, there could be potential residual confounding factors (due to the design) which may influence the relationship between adiposity markers and cardiometabolic risk (it should be accounted as a limitation).

Answer: We added this limitation in the limitations section :

“Another important limitation is that this was a cross-sectional study, which does not allow for the identification of causal relationships. Similarly, there could be potential residual confounding, and due to variability in W-HtR across different populations, our findings may not be generalizable until they are replicated in other populations.”

Comment 4: The authors stated regarding the ethics: “The study applied the guidelines of the Declaration of Helsinki and Resolution 8430 of the Colombian Ministry of Health, which state that this is a risk-free investigation. The participating institutions were endorsed, and the project was endorsed by the ethic committee of Metrosalud. The information was collected based on medical records”. Probably an approval number could be required. Also, a statement is required regarding the informed consent of the patients. Such statement like “this is a risk-free investigation” should be avoided.

Answer: The following information regarding the ethics committee was added

The project was approved by the Metrosalud Ethics Committee during the session on October 12, 2021, under communication code 1400/3.

On the other hand, according to Resolution 8430 of 1993 issued by the Ministry of Health of Colombia, Article 11 classifies research based on the review of medical records (such as ours) under the category of risk-free research. Due to this nature, informed consent was not obtained, as the work involved reviewing medical records of patients treated at the hospital during the year 2019.

Comment 5: The limitation section should be updated (it states almost only strengths, although the design of the study was cross-sectional and inferential analysis could be misleading)

Answer: As we mentioned above, the section of weakness was updated

Comment 6: Inclusion and exclusion criteria should be better defined in the methods.

Answer: We provide a detailed description of the inclusion and exclusion criteria for participants as supplementary material, as follows:

Table S1. Definition of potential extreme/implausible values for anthropometric and biochemical variables

Values

Glucose (mg/dl)

<30 or >600

 Triglycerides (mg/dl)

<10 or >1.000

HDL-C (mg/dl)

<10 or >118

SBP (mm / Hg)

<80 or >280

DBP (mm/Hg)

<50 or >195

Waist (cm)

<50 or >198

BMI (Kg/m2)

>70

SBP=Systolic Blood Pressure; DBP=Diastolic Blood Pressure; C=Cholesterol.

Table S2. Identification of the study sample

Initial sample

Exclusion criteria

n excluded

Remaining sample

69,883

Missing values for

anthropometric and biochemical variables

19,505

50,378

50,378

Potential extreme/implausible values for

anthropometric and biochemical variables

2,589

47,789

47,789

Diagnostic of  any of the following diseases: diabetes, kidney disease, cerebrovascular and cardiovascular disease

10,231

37,558

37,558

Missing values for covariates of age, sex, education level, ethnicity, area of residence, marital status, alcohol consumption, and smoking

8,322

29,236

(Final sample)

Reviewer 2 Report

Comments and Suggestions for Authors

The introduction provides a solid background on cardiometabolic risk factors and their relationship with adiposity markers. It discusses the significance of BMI, waist circumference (WC), and waist-to-height ratio (W-HtR), citing previous studies. However, while references to global research are included, there is limited discussion of prior studies conducted in similar Latin American populations. Expanding on regional studies would improve contextual relevance. The cross-sectional study design is appropriate for examining associations between adiposity markers and cardiometabolic risk. The large sample size (n=29,236) strengthens the reliability of the results. However, being cross-sectional, it does not allow causal inferences. The methods section is detailed and describes how data were collected, including the exclusion criteria, measurement techniques, and statistical analyses. However, a more explicit discussion of how missing data were handled and how potential confounders were selected would enhance clarity. The results are well-structured with clear tables and figures. The logistic regression models and cubic spline analyses are well explained. However, the readability of some sections could be improved by summarizing key findings before presenting detailed statistical outputs. The conclusions accurately reflect the findings, emphasizing the strong predictive role of W-HtR in cardiometabolic risk. However, the authors suggest W-HtR should be the preferred screening tool without adequately acknowledging potential limitations such as variability across different populations. 

I have minor suggestions for Authors. Please improve the clarity of the introduction by briefly discussing prior research conducted in similar Latin American populations. This would provide better context for the study. Clarify methodological details by regarding how missing data were handled and how confounders were selected in the statistical models. Improve the readability of the results section by summarizing key findings before presenting statistical details. This will help non-expert readers interpret the findings more easily. Acknowledge the limitations of the study more explicitly, particularly regarding its cross-sectional nature, potential measurement biases, and generalizability to populations outside of Medellín, Colombia. Justify the recommendation to prioritize W-HtR over BMI and WC with additional discussion on whether this applies universally or mainly to populations with specific characteristics.

Comments on the Quality of English Language

The manuscript contains grammatical errors, awkward phrasing, and redundancies. Certain sentences are overly complex, making them difficult to understand.

Many sentences contain grammatical errors, missing articles, or awkward phrasing.  
for example: "The full cardiometabolic risk was defined as having all three kind of components." should sound more something like "Full cardiometabolic risk was defined as the presence of all three components."

Some sentences are unnecessarily long or convoluted, reducing clarity.  
for example: "The W-HtR-full cardiometabolic risk association appears to start from a value of W-HtR around 0.6, while full cardiometabolic risk start to be confidently associated with WC from values of this marker close to 100 cm." should sound more something like "The association between W-HtR and full cardiometabolic risk becomes evident at W-HtR values of approximately 0.6. Similarly, full cardiometabolic risk is strongly associated with WC values around 100 cm."

Some phrases are repetitive and could be streamlined.  
for example: "This suggests that when individuals have a W-HtR of around 0.6 or higher, they are more likely to exhibit an increased risk of developing cardiometabolic conditions." should sound more something like "A W-HtR of 0.6 or higher is associated with an increased risk of cardiometabolic conditions."

Some sections switch between past and present tense inappropriately.  
for example: "In our study, we found that the W-HtR adiposity marker most strongly correlated with cardiometabolic risk." should sound more something like "In our study, W-HtR was the adiposity marker most strongly correlated with cardiometabolic risk."

The manuscript often omits necessary articles or misuses them.  
for example: "Waist-to-height ratio has been increasingly recognized as superior screening tool for assessing cardiometabolic risk." should be "The waist-to-height ratio has been increasingly recognized as a superior screening tool for assessing cardiometabolic risk."  

A thorough professional English revision is strongly recommended to improve readability, grammar, and clarity. While the scientific content is strong, refining the language will enhance the manuscript’s impact and ensure it is accessible to a broader audience.

Author Response

(The authors gave the same response as above.)

Reviewer 3 Report

Comments and Suggestions for Authors

The article investigates the association between different adiposity markers (waist-to-height ratio, waist circumference, and body mass index) and full cardiometabolic risk in an adult population from Medellin, Colombia. The study is relevant given the increasing burden of metabolic disorders and cardiovascular diseases in Latin America. The authors utilize a large dataset (n = 29,236) from a chronic disease prevention program. The novelty of the study lies in its focus on a South American cohort and the examination of W-HtR as a potentially superior predictor of cardiometabolic risk. However, some issued should be addressed:

1) The authors concluded: “In this large middle-to-older-aged cohort, W-HtR was the strongest adiposity marker linked to cardiometabolic risk, with a more linear associa-tion pattern”. However, given the cross-sectional design of the study, results on such association or prediction could be misleading (the study design does not allow for causal inference);

2) Given the fact that it represents a novel paper on cardiovascular and metabolic factors, it would be useful to discuss the (relative) new cardiovascular-kidney-metabolic (CKM) syndrome;

3) Also, there could be potential residual confounding factors (due to the design) which may influence the relationship between adiposity markers and cardiometabolic risk (it should be accounted as a limitation);

4) The authors stated regarding the ethics: “The study applied the guidelines of the Declaration of Helsinki and Resolution 8430 of the Colombian Ministry of Health, which state that this is a risk-free investigation. The participating institutions were endorsed, and the project was endorsed by the ethic committee of Metrosalud. The information was collected based on medical records”. Probably an approval number could be required. Also, a statement is required regarding the informed consent of the patients. Such statement like “this is a risk-free investigation” should be avoided.

5) The limitation section should be updated (it states almost only strengths, although the design of the study was cross-sectional and inferential analysis could be misleading)

6) Inclusion and exclusion criteria should be better defined in the methods.

Comments on the Quality of English Language

Minor English edditing required.

Author Response

(The authors gave the same response as above.)

Round 2

Reviewer 1 Report

Comments and Suggestions for Authors

I have no comments, my suggestions were taken into account.